# Addressing the Mental Health Challenges of Refugees—A Regional Network-Based Approach in Middle Hesse Germany

**DOI:** 10.3390/ijerph192013436

**Published:** 2022-10-18

**Authors:** Bernd Hanewald, Michael Knipper, Janneke Daub, Saskia Ebert, Christina Bogdanski, Laura Hinder, Mila Hall, Daniel Berthold, Markus Stingl

**Affiliations:** 1Center for Psychiatry and Psychotherapy, University of Giessen, 35392 Giessen, Germany; 2Institute for the History of Medicine, University of Giessen, 35392 Giessen, Germany; 3Refugee Law Clinic Giessen, Public and European Law at the University of Giessen Law School, 35394 Giessen, Germany; 4Center for Psychosocial Counselling, 35390 Giessen, Germany; 5Research Network on Migration and Human Rights, Public and European Law at the University of Giessen Law School, 35394 Giessen, Germany; 6Department for Medical Oncology and Palliative Care, University of Giessen, 35392 Giessen, Germany

**Keywords:** refugees, trauma network, human rights, syndemic care, trauma

## Abstract

Refugees constitute a vulnerable group with an increased risk of developing trauma-related disorders. From a clinician’s integrative perspective, navigating the detrimental impact of the social, economic, structural, and political factors on the mental health of refugees is a daily experience. Therefore, a collective effort must be made to reduce health inequities. The authors developed a treatment concept which provides broader care structures within a scientific practitioner’s approach. The resulting “Trauma Network” addresses the structural challenges for refugees in Middle Hesse. Accompanying research provided a sound basis for further discussions with policy-makers to improve the situation for refugees in the short- and long-term.

## 1. Introduction

The potentially harmful effects of social, political, and ecological factors on the health of individuals and populations urge for joint efforts within and beyond health services [1,2,3]. This is especially relevant in the case of refugees and migrants, whose access to health care and to the social, structural, and economic preconditions of mental and physical health are often hampered by politically and legally defined barriers and constraints [4,5,6]. Given how intertwined these factors are with health and wellbeing outcomes, health services need to find new ways for doing their part in reducing health inequities, even in industrialized countries. As a useful proposal for better tackling and responding to the complexity of the mental health challenges of refugees, Willen et al. [7] suggest that “a combined syndemics/health and human rights approach can help clinicians better connect the dots among upstream determinants, complex clinical realities and social justice obligations, and furthermore, clarify both the value and feasibility of doing so” (pp. 974). The term “syndemics” refers to a model for addressing health and disease in research and clinical practice in a way that systematically draws attention on both the interactions between different disease entities (comorbidities and concurring epidemics) and the driving factors for better or worse health outcomes in the social environment [2,8,9]. Addressing and alleviating the co-occurrence and mutually reinforcing influence of more than one disease in relation to social, cultural, legal, and political factors is the major aim of the syndemics perspective. By relating this to a comprehensive perspective on health as a human right, and that perceives the relevant social, legal, and political factors as “underlying determinants of health” [10] that offer critical levers to advance equity in health, Willen et al. [7] provide a practical way from analysis to action.

From the perspective of clinicians and clinical psychologists, the often detrimental effects of social, economic, structural, and political factors on the mental health and wellbeing of refugee patients is particularly clear. Refugees and asylum-seekers are known to constitute a highly vulnerable group with an increased risk of developing mental illnesses. According to international studies, 30 to 70% of refugees are suffering from trauma-related disorders such as post-traumatic stress disorders (PTSD), depression, and anxiety [11,12,13,14,15,16,17,18]. The high prevalence of mental illnesses among refugees is exacerbated by limited medical treatment options in the psychiatric field, a problem that persists for various reasons. At the same time, mental illnesses represent a serious obstacle to sociocultural integration and can have considerable financial and social consequences [19,20]. Often, due to structural restrictions (e.g., limited duration of inpatient treatment, language barriers, transcultural difficulties, and long waiting times for outpatient psychotherapy), exclusively psychopharmacological interventions are used when there are insufficient resources to provide therapeutic stabilization. The trauma-related disorders can stem from witnessing various types of war-time experiences from the perspective of a civilian or military personnel. Examples of these types of experiences can include, but are not limited to, witnessing extreme violence such as torture and gender-based violence, as well as direct combat [21]. In therapy, it is essential for members of military personnel to come to terms with the fact that they simultaneously embodied both the victim and the perpetrator roles. Due to the fact that the bureaucratic asylum-seeking process is extremely complex and often takes a very long time, the lives of refugees can be immensely destabilized as they wait to be granted official refugee status. This destabilization can manifest in many different ways, including living under precarious conditions, loneliness, helplessness, the lack of fulfilling and meaningful daily activities, grief, the loss of social structures, social identity, and/or family [22,23,24]. All of these factors can accumulate and reciprocally amplify each other over time, further complicating the psychological care of refugees and asylum-seekers. Moreover, the often protracted situation of insecurity regarding the options to stay safe, to being granted asylum or not, that in many cases prevail for unpredictable time periods between weeks and years, seriously undermine the feasibility of psychotherapy. Aside from that, the social aspects of the migration process add a further layer of complexity. In many cases, the journey to flee the country of origin has been financed by family members despite their poor economic situation. The arrangements often include the expectation that the individual who left will then financially support those who stayed at home. In other cases, families expect to reunify in the allegedly safe environment of the destination country in the future. If these expectations cannot be met (e.g., to provide support to children, parents, or spouses who have remained in precarious living situations), strong feelings of guilt can result on the part of the refugee [25,26]. In addition, insufficient knowledge of basic asylum law as well as the outright denial of the legal system’s influence on refugee mental health further complicates support and mental health care [27]. Language barriers and challenges related to working with interpreters can be difficult for both the asylum-seekers as well as those providing treatment [28]. Discussions related to who or which institutions are responsible for covering the costs of a necessary inpatient stay also complicate the provision of care. While cultural aspects should not be overlooked in the treatment of refugees, too much emphasis on the presumed “culture” can produce one-sided stereotypes and jeopardize an empathic, individualized approach to treatment [29,30].

In the experience of the authors, however, one of the most relevant upstream factors, and thus a critical determinant of refugee’s mental health, is legal status [31,32]. This assertion is corroborated by legal research, which confirms that securing a resident status is essential for noncitizens to gain access to social systems in host societies [27]. A lack of a sense of security due to uncertainties surrounding their resident status hinders refugees in accessing and accepting health care. For example, existential fears of being deported interfere with psychotherapeutic progress and can lead to resignation, hopelessness, and even acute suicidal tendencies [31]. Standard psychotherapeutic approaches that ignore the legal dimension not only fail to address the real emotional needs of refugees, but also fall short in meeting their practical needs. In order to provide truly syndemic care for refugees, legal counseling must be considered an additional essential resource. This is especially relevant since legal counseling and psychotherapy can help address different kinds of fears refugees may be facing. Handling legal concerns may help to manage fears of an impending deportation, thereby helping to establish refugees’ feelings of safety and belongingness in their host country. Once these immediate and tangible concerns are settled and a secure base is established, the hard work of restoring an internal sense of security can be facilitated through psychotherapy. For patients suffering persistently from the external, disease-aggravating stressors mentioned above, the diagnosis of PTSD does not entirely accurately reflect the presenting problem: there is no true “post”-traumatic situation for many refugees. Since the traumatic situation, many refugees are immediately exposed to several other chronic stressors, be it the escape itself, navigating a new country and culture, or sustained fears of being deported to the country where the trauma originated.

The consequences of trauma, but also structural conditions in the care of refugees with, among other things, uprooting, poor social integration, subjectively perceived helplessness, and limited access to health care, as well as legal conditions with often prolonged “legal limbo” and uncertainties regarding the future with uncertain residence status, can negatively influence the coping resources of refugees as well as the process of recovery and healing [33]. Cognitive distortions and reduced self-efficacy after trauma may lead to impaired adaptive skills, impaired ability to regulate emotions, and reduced ability to recover from trauma-related limitations [34,35]. In particular, the inverse relationship between self-efficacy and trauma sequelae has been repeatedly demonstrated [35,36,37].

The aforementioned factors must be understood as ongoing stressors that severely impede progress in psychotherapeutic treatment, exacerbating existing psychological problems, or triggering the development of other complaints. From a human rights perspective, it is therefore essential to provide care to refugees using a multidisciplinary framework, one not only focused on psychiatric–psychotherapeutic treatment, in order to ensure that their comprehensive needs are being met. Consequentially, the authors would like to present the development of a scientist–practitioner model for preventing trauma and decreasing its negative effects in refugees from a clinician’s perspective. This effort was made possible and expanded by the outstanding commitment of many engaged people and should ideally serve as a proposal and source of inspiration for building systems that adequately address mental illness among refugees.

## 2. The Scientist–Practitioner Approach

Giessen, a small university town located in central Germany, has a long history of taking in displaced people and refugees over the last three quarters of a century. In 1946, and as a result of the Second World War, the “Giessen government transit camp” was founded in the US-occupied zone in Germany to accommodate displaced people from the Soviet zone. After the violent suppression of the uprising on 17 June 1953 in Eastern Germany (the former German Democratic Republic, GDR) and the building of the wall, Giessen and its centrally located transit camp became a symbol of freedom for nearly 900,000 people who fled or were relocated from East to West Germany. Approximately 120,000 of them arrived within the first year following German reunification in 1990. Since 1993, the Giessen initial reception center is the main location for asylum applicants arriving in the state of Hesse. While around 17,000 asylum-seekers reached Giessen in 2014, this number increased to almost 80,000 people in 2015 and has dropped again to 61,075 people since 2016 to date (according to the Press Office of the Giessen Regional Council, October 2020). Given this formative tradition and the current high demand, a group of committed people began collaborating across various sectors to address the dearth of psychosocial resources for refugees. Following numerous personal contacts, it became clear that the motivation to help those in need was shared by supporters across different professions and positions. To bring these supporters together, the “Trauma-Network Middle Hesse” was founded. Soon, regular consultations began taking place (for a detailed description see below). At the beginning, these consultations consisted of members exchanging diverse viewpoints from their respective fields of expertise on selected cases of refugees in need of help. It became clear that providing adequate help in the long run for these kinds of cases would need to be a consistent, collaborative, and cross-sectoral undertaking. The authors additionally recognized that all action would be related to or at times contingent on specific medical, political, and legal context factors. Taken together, it had to be determined that the intention to provide support immediately exclusively through psychotherapy for potentially traumatized refugees was neither sufficient nor did it approach the potential of what could be possible. How could the gap between the high need for support on the part of the refugees and the inadequate offers of support be closed? What was needed in addition to the engagement of a few committed individuals? Thinking about these core questions in the network, it became clear that good comprehensive care could only succeed if the groundwork had already been laid in the form of appropriate and reliable multisectoral offers. The authors came to the conclusion that two paths should be pursued further, from which solutions and success in the short and longer term could be expected: First, at the practitioner level, it was decided to continue the procedure of joint care with consistent advancements through direct and rapid exchanges amongst one another. Through this process, existing alliances and collaborations within the network were deepened and new ones were established so that the sharing of knowledge and access to expertise became more and more routine. Secondly, the authors decided to use their scientific expertise to evaluate and improve operations and strategies. Using scientific methodologies, the authors aimed to provide empirical data to help inform policies to support refugees in the short- and long-term (see section “Center for Psychosocial Counselling”).

In summary, by applying a scientist–practitioner lens, the authors aim to address the urgent unanswered questions related to providing support to refugees on a local level. The aim is to show how to establish an actionable framework for comprehensive refugee care by using and expanding existing local structures. This paper presents the current status of the framework as it exists in Giessen and the steps that were taken in the process of its evolution. The authors hope to demonstrate how collaborative synergies can be used in service of refugees’ well-being. For this reason, the various structures that make up the Giessen Trauma Network will be presented one by one from a practical point of view, followed by a description of the scientific collaborations, which will then be contextualized in the broader body of research literature.

## 3. Establishing the Practitioner Nexus

In an effort to promote structural change and raise awareness of the political, social, legal, and cultural determinants of mental health among refugees, the regional multidisciplinary Trauma Network was established in 2012. This network is an association of people actively contributing to and advocating for the improvement of psychosocial care for refugees in the Hesse region of Germany. Its professional diversity provides the opportunity to offer support on a variety of levels. The network is open for participation and membership is not formally tied to any obligations or conditions. It can be stated that the voluntary and altruistic idea of additional involvement in a trauma network is beneficial, not least to the supporters of the network. In several studies it could be shown that altruistic behavior and voluntary, honorary engagement leads to the improved well-being of the engaged themselves [38,39,40,41]. In our concrete example, this leads to the sustainable establishment of a network, in the meantime over more than 10 years, with the largely stable cooperation of the different actors. Or, as the patient in the case study described below put it succinctly: “Don’t take charity if you can give charity”.

There are Trauma-Network meetings about four to six times a year, forums that can be used for mutual exchange, advice, and further training in the rooms of the Clinic for Psychiatry and Psychotherapy of the University Hospital in Giessen or one of the local training institutes for psychotherapy. The exact content of meetings and trainings is adapted in order to best serve the attendees’ needs. The network operates a homepage in order to increase awareness and visibility to the general public as well. Members of the Trauma Network include: medical doctors and psychologists from various clinics in the area (e.g., the Regional Clinic for Psychiatry and Psychotherapy Giessen, the Clinic for Psychosomatics and Psychotherapy, and the Clinic for Psychiatry and Psychotherapy at the University Hospital in Giessen), pastors, medical and psychological psychotherapy residents, interpreters and language/cultural mediators (some of whom have their own refugee experiences), volunteers, members of the Refugee Law Clinic Giessen, asylum procedure counselors from the Protestant Church, charity employees, members of three Centers for Psychosocial Counselling in Hesse, as well as members of Medinetz. The aforementioned participants provide their support to the network according to their respective expertise. In the following, the authors introduce some of its many important contributors and their activities.

Within the *Clinic for Psychiatry and Psychotherapy* at the University Hospital in Giessen, there are two specialized units where refugees can obtain psychiatric and/or psychotherapeutic help: First, the Trauma Therapy Center offers outpatient help for traumatized people. As mentioned previously, the Trauma Therapy Center also acts as an intermediary between inpatient and outpatient treatment and relays relevant information on to other network partners. Second, an integrative treatment plan for the psychiatric care of refugees was implemented in the clinic. In designing this plan, the authors relied on suggestions for “syndemic care” as put forth by Singer, Mendenhall, Willen and colleagues in the Lancet Series on Syndemics in 2017 [2,7,42]: a systematic inclusion of legal, social, structural, and other syndemic factors into standard evidence based psychiatric care. As part of this syndemic care plan, collaborations with the University of Giessen Law School (“Refugee Law Clinic”), lawyers, and Asylum Procedure Counseling of the Protestant Church were built to provide legal resources, mutual support, and counseling as an integrated component of psychiatric routine care and to consider whether an official medical opinion could be of further use in the asylum procedure. The Giessen inpatient treatment concept for refugees is a practical model for the care of traumatized refugees in inpatient psychiatric structures which includes multiprofessional and interdisciplinary cooperation and networking, and which has been described previously [43,44]. Guidelines have been drawn up for individual therapeutic interviews, which follow three lines of inquiry: (1) individual therapeutic focus, (2) learning of stabilization techniques, and (3) consideration of legal aspects (if the patient is still in the asylum procedure and does not have secure resident status).

In clinical care for migrant patients, addressing culture and perceived cultural differences is crucial and challenging. Cultural stereotypes and essentialist beliefs about culture represent a serious barrier for true patient-centered care, since they hinder opportunities to more deeply engage with the patient’s unique lived experiences (especially when the treating clinicians come from different cultural backgrounds) [45]. Furthermore, it is essential to consider the impact of institutional cultures in hospitals and health facilities (e.g., regarding conventional ways of communicating, organizing the patient’s daily routine, and addressing both the somatic and psychosocial dimensions of health), as is acknowledging and actively working to dismantle racism and discrimination at the structural and individual level. In order to sustainably strengthen cultural competence and sensitivity among staff, a concept of a culturally sensitive supervision for the interprofessional team of the ward was developed and moderated by a physician and medical anthropologist. The moderator’s main role was to facilitate the discussion and to challenge the narratives and interpretations with the following two questions: (1) What do “culture” and “migration” mean concretely in this particular case? (2) What are the major concerns of the patient, and how does this compare to the team’s expectations and priorities, as well as to the social reality and lived experiences of the individual? In many instances, the shared reflection process revealed that “culture” played rather a secondary role, while very concrete experiences, concerns, and problems were at center stage for the patient. Culture played an important role in that it was often the source of how these experiences, concerns, and problems were expressed and acted upon. The group shifted its focus from talking about culture to talking with and reflecting about individual patients in a more holistic way, using culture as a perspective through which specific questions regarding experiences, values, habits, and social relations could be addressed, both on the patients’ side as well as the team’s.

The *Refugee Law Clinic (RLC)* is a hands-on, interdisciplinary training program in refugee law. The goal is to impart knowledge of German, European, and international refugee law to students during their studies and to deepen this knowledge through practical application by counseling refugees. For one, it enables students to develop their competencies, and additionally, it expands access to legal advice and support for people seeking asylum. The education begins with a theoretical part: the students first listen to lectures on German, European, and international refugee law to create a baseline of legal knowledge. At the same time, the first interdisciplinary basics are acquired in group project meetings. From the very beginning, the students learn about the existing structures in the field of refugee and migration issues that are locally available. The students grapple with topics such as racism and discrimination, gain insights into the psychological support of refugees, and the social and labor law framework conditions for refugees in Germany. The networking with other disciplines is of great importance to the activities of the RLC Giessen. The collaborations with the Clinic for Psychiatry and Psychotherapy at the University Hospital in Giessen and the Giessen Medinetz are particularly essential and are not to be missed. In cooperation with the Clinic for Psychiatry and Psychotherapy, a psychological supervision is offered to the students in which they can reflect on their own role and experience in counseling refugees [46]. The students complete a first internship with a lawyer specializing in refugee law, a migration law authority, or an NGO actively working with refugees and continue in consulting at the RLC Giessen, which is offered in one of two ways. First, the members of the RLC may advise people in the asylum procedure who reach out with a specific question. Alternatively, the RLC also offers so-called information evenings, which are essentially consultations for groups of people. The latter usually takes place in the “Reception Center”, a Refugee Center for Registrations in Giessen. These consultations provide general information about the asylum procedure, the rights and obligations of the applicants, and represent the first contact to a counseling center for many refugees, thus enabling them to obtain legal support. The students are encouraged to attend further training courses on a regular basis, thereby continuously consolidating and expanding their knowledge. The students gain access to academic and practical work outside the university context and have the opportunity to independently get in touch with national and international migration law networks. However, the RLC Giessen is also present outside the university context. The students are allowed to teach the basics of migration law to people from other disciplines and are thus contact people for legal questions for different actors in refugee work. The aim is to spread basic knowledge of refugee law and to increase the legal competence of people working with refugees. The RLC Giessen also regularly participates in events organized by various networks, including nonlegal ones, to maintain an exchange of views on the various aspects of refugee work and to learn nonjudicial skills.

*Medinetz* is a network of volunteer medical students and health professionals with active groups in Giessen and many other German cities. It organizes medical assistance for refugees without health insurance and/or without valid identification documents or a valid resident status, because they are largely excluded from medical care.

Outpatient treatment of refugees is offered by resident therapists as well as the Outpatient Clinic of the Institute for Psychotherapy and Psychoanalysis Giessen and the Behavioral Therapy Outpatient Clinic at the University of Giessen. The latter two institutions are also educational institutes for trainee psychotherapists. Following an inpatient stay, the Trauma-Network helps to ensure that receiving outpatient psychotherapy can be started as soon as possible. In addition, the Trauma Network also fosters scientific collaboration, since some of its members are affiliated with the complementary Research Network on Migration and Human Rights (see below).

In regular exchanges between members of the Trauma Network and policy-makers of the Ministry of Social Affairs, the need for the establishment of a *Center for Psychosocial Counselling* could be presented and justified with epidemiological data collected by the authors. Subsequently, in 2017, a founding association was established by several members of the Trauma Network. As a main outcome, it receives annual funding from the Hessian Ministry for Social Affairs and Migration and currently operates the Center for Psychosocial Counselling Middle Hesse (https://www.psz-mittelhessen.de (accessed on 10 September 2022)) with a team of psychotherapists, social workers, administrative staff, and cultural mediators. This interdisciplinary team offers psychological assessment, psychoeducation, skills for stress and anxiety management, psychological crisis intervention, and referrals to further psychiatric and psychotherapeutic treatments if needed. The services are available to refugees living in the initial reception centers but can also be accessed by those in private accommodations in Middle Hesse. Counselling sessions are mostly held within an individual context, but group sessions are also available. Psychosocial counseling sessions are translated with the help of cultural mediators that are trained to provide precise and professional translations, both of language and of culture, in this specific work context. The professional involvement of cultural mediators with their own migration backgrounds and associated lived experiences adds an important layer of additional expertise for the practitioners. The Center for Psychosocial Counselling also offers supervision for cultural mediators, anonymous expert advice for cases within other institutions, and workshops on mental illnesses related to trauma. To strengthen existing networks and optimize psychosocial care for refugees more broadly, the Center for Psychosocial Counselling cooperates with institutions and associations all over Hesse. The founding association also reports psychosocial issues relevant for the improved mental health of refugees back to the Hessian Ministry for Social Affairs and Migration.

Figure 1 provides an overview of the different actors in the network. Particularly in the treatment of trauma sequelae, cross-collaborations between, for example, practitioners, researchers, and advocacy groups have been shown to offer more benefits than the usual translation of science into clinical practice [33,47,48,49,50].

To the best of our knowledge, there are superordinate umbrella organizations in Germany, some of which operate on a nationwide basis, which, for example, network individual psychosocial care centers with one another (e.g., The German Association of Psychosocial Centres for Refugees and Victims of Torture (BAfF), https://www.baff-zentren.org/english/ (accessed on 10 September 2022)) and carry out public relations work. At the local level, however, we are not aware of any similarly differentiated and multiprofessional participatory networks.

## 4. The Scientific Part: Research

As described above, the Center for Psychiatry and Psychotherapy at the University of Giessen decided not only to promote and bundle regional activities in refugee treatment and care, but also to support, through research activities, the process of building up structures and treating refugees from the very beginning. In addition, the authors draw on the scientific expertise of various members of the network from various disciplines. To develop and promote suitable offers for refugees, first and foremost the demand must be determined and measures to identify those in need of help must be implemented. In this sense, the European Parliament and the European Council stipulation from 2013 (European Parliament, European Council, 2013) [51] made it obligatory for member states to provide appropriate medical care (including psychological care) to so-called vulnerable refugees. In their classification, vulnerable refugees included infants, people with disabilities, pregnant women, people with severe physical illness or mental disorders, as well as people who experienced torture, rape, and other interpersonal (sexual) violence. However, the implementation of the directive has been inadequate so far. The Trauma-Network has taken up this topic and conducted funded/nonfunded projects and research studies.

### 4.1. Identifying Common Trauma-Related Disorders

As concrete measures have yet to be taken by the responsible authorities to identify refugees in need of protection, the authors have considered how this could be implemented in practice. It was decided to investigate the feasibility of identifying the most common trauma-related disorders as validly as possible, and thus proposed a two-step procedure that includes a screening questionnaire and, if positive, further diagnostic procedures. For the first step, screening, the Refugee Health Screener-15 (RHS-15) by Hollifield et al. [52], which is culturally sensitive and records the most common complaints in different affected refugee subgroups, was selected. As the RHS-15 has shown good cross-cultural validity and reliability in different refugee groups, the authors examined its applicability on-site using further scientific quality criteria such as standardization, test fairness, reasonableness, susceptibility, transparency, acceptance, external design, utility, and economy. The authors demonstrated that the RHS-15 was a valid, reliable, and efficient first screening instrument, and therefore very suitable for broad use. A further finding of this study showed that a large proportion of the examined refugees suffered from serious psychological symptoms, indicating a high prevalence of trauma-related disorders [15]. This finding was moderated by the time of screening as well as the duration of stay in Germany. Evidence of trauma-related disorders were found among 65.9% of refugees living in reception facilities immediately after arrival (n = 125) and more than 80% of refugees living in community shelters (n = 116).

In another project, and in cooperation with a general practitioner who was responsible for the initial medical examination of unaccompanied refugee minors (URM), the prevalence and patterns of trauma-related disorders in 561 URM who spoke four different groups of languages (Arabic, Farsi, Somali, and Tigrinya) was studied using a sequential mixed methods approach [53]. Due to their young age, developmental status, and insufficient coping strategies, URMs represent one of the most vulnerable refugee groups [54,55,56,57,58,59]. It could be demonstrated that the frequency of mental health problems varies between different URM subgroups. A higher burden was observed among older and male URMs. Positive screening values ranged between 42.9% and 62.3%, with variations between different regions of origin. Somali and Farsi speakers, the latter mostly coming from Afghanistan, reported much more mental strain and had higher positive screening results in the RHS-15 compared to Arabic, mostly from Syria, and Tigrinya speakers from Eritrea. The observed group differences can be understood as expressions of syndemic vulnerability caused by upstream factors in the country of origin, such as political conditions, rivalries and blood feuds between clans, war, and impeded access to educational institutions. Somalia has had a long-lasting history of warfare and combat, especially in the southern regions. The URMs and the generation before had to live under unstable and traumatic living conditions for many years, which may help to account for an insufficient development of resilience factors within the next generation. The same holds true for Farsi-speaking URMs. Decades of warfare within the country have increased the risk for transgenerational traumatization [60,61]. Frequently, there are no opportunities for these children to regularly attend schools. In 2021, in Afghanistan, 37.3% of people aged 15 and above were able to read and write with understanding a short simple statement about their everyday life [62]. Furthermore, in Germany, at that time adult asylum-seekers from Afghanistan had a fairly small chance of success in the asylum procedure and/or had to endure a great deal of uncertainty until a final decision concerning their asylum application had been made. In addition to the individual traumatic experiences, many other related variables additionally compound migration stress. These include, but are not limited to, irregular or entirely missed years of education due to conflict- or war-related variables, interruption of education more broadly during escape, and increased demands due to lower language skills in asylum countries [63,64]. The participating unaccompanied minor refugees were all considered to be a particularly vulnerable group due to their age and the absence of an accompanying adult. The adult refugees were also considered to be associated with an increased vulnerability to mental illness, particularly trauma-related disorders, due to their experience of flight. The extent to which this resulted in indications of a special need for protection was the subject of the investigations.

Additionally, the Clinic for Psychiatry and Psychotherapy at the University Hospital in Giessen is involved in two collaborative projects targeting refugee support and treatment. The first project, in cooperation with another regional Clinic for Psychiatry and Psychotherapy (Vitos Clinic Giessen) and the Clinic for Family Psychosomatics at the University Hospital in Giessen, focuses on the psychological stress of children in families that fled to Germany. The authors are trying to understand the connections with the parents’ own escape experiences and possible psychological illnesses of the parents. The results of this ongoing trial will then be compared with a large sample of children whose parents did not experience escape/fleeing, but have a pre-existing psychiatric disorder, so to get more detailed information about the unique needs of refugees’ children in Germany. The long-term goal is to use this information to further develop appropriate treatment approaches for these children, as many treatments for adults are already available and empirically backed. In another project, the Clinic for Psychiatry and Psychotherapy at the University Hospital in Giessen is participating in the clinical evaluation of a portable communication system which can be used on-site during psychiatric diagnosis for Arabic-speaking people. The aim of this joint project, funded by the German Federal Ministry of Education and Research, is to minimize obstacles in providing mental health care for refugees: RELATER—Removing Language Barriers in Treating Refugees (German Clinical Trials Register—ID: DRKS00024090).

All of the above studies were approved by the local ethics board. In the studies presented, care was taken to minimize the burden on participants and Good Clinical Practice guidelines were followed. Only staff trained in dealing with refugees and trauma sequelae were used in the implementation. In this process, participants were informed in detail that the interview was voluntary and that there would be no consequences if they refused to participate or terminated the interview prematurely. In particular, if they were in a treatment setting, they were informed that refusing the survey would not affect their treatment. When recording indications of the presence of a trauma sequelae disorder, potentially traumatic events were not asked about, but only indications at the symptom level—e.g., restlessness, startle, nightmares, or pain—were recorded.

Representatives of the Clinic for Psychiatry and Psychotherapy at the University Hospital in Giessen are also members of the interfaculty research network which tries to promote research under a comprehensive human rights-based approach, which will be explained further in the next subsection.

In summary, the authors demonstrated that the RHS-15 is a useful, economically valid, and efficient tool to screen individuals for trauma-related mental health problems in a first step. In a second step, positive screenings should be followed by further diagnostic measures. This structured two-step diagnostic approach allows to fill the gap between legal requirements (EU directive) and individual needs. The alarming prevalence of mental health complaints in URMs highlights the need for early symptom detection and timely psychosocial interventions to prevent chronic courses of disease [65] and severe behavioral problems [59]. Suffering from mental disorders at a vulnerable stage of life such as adolescence can give rise to severe development difficulties, resulting in obstacles hindering social integration, problems of bonding capacity and competency in relationships, as well as chronic courses of disease. The provision of early treatment helps to prevent the aggravation and continuation of emotional problems [66]. Therefore, the early detection of URMs at risk for developing mental impairments is an essential preventive measure to avoid later undesirable development in connection with higher costs and poorer integration. In view of this, the early detection of mental health problems after arrival is especially important for both the URM and the host county. It enables adequate support during the vulnerable transition period from adolescence to adulthood as well as a successful transition to an integrated life in a new culture and society [67]. The results of the above-mentioned studies were made available to the responsible authorities and formed the basis for a discussion about the insufficient care structures for refugees on-site. Together with other actors from all over the country, the authors were able to advocate for the rights of refugees, to address the gap in care structures available to them, and to generally initiate further reflection amongst political decision-makers about the need to prioritize refugees’ needs. As one of the first positive outcomes of this research-informed advocacy process, four centers for psychosocial counseling were established in Hesse, which offer psychosocial support for refugees, mainly cofinanced by the state of Hesse (see above).

### 4.2. The Research Network on Migration and Human Rights

The Research Network on Migration and Human Rights (FMM, https://www.migrationundmenschenrechte.de/de/topic/542.english-description.html (accessed on 10 September 2022) exists complementary to the Trauma-Network. Members are researchers from various academic fields at the University of Giessen, coordinated by the Chair of Public and European Law at University of Giessen Law School. While the Trauma-Network focuses mainly on the treatment and care of refugees, the FMM mainly deals with issues concerning refugees from a human rights perspective. Therefore, the FMM requires contributions from all academic fields. The research network was founded by members from the Schools of Law, Educational Sciences, Sociology, and Health in 2014. It now includes more than 50 academics from various fields such as law, history, sociology, political sciences, peace and conflict studies, educational sciences, theology (Christian and Muslim), and health. In addition to its involvement in multiple research projects, it also offers a teaching program on migration and human rights that links disciplines, perspectives, theory, and practice. The group meets regularly for joint workshops where current research activities or external inputs are discussed.

### 4.3. Case Example

Mr. S., a 35-year-old man from Mali, was presented to the trauma outpatient clinic by Medinetz. He was threatened by Islamist groups in Mali because he has two nonmarital children. During his escape, he was detained and tortured for over 8 months in Libya. He entered Germany via Italy and became homeless without valid documents. He was hospitalized due to paranoid psychosis and post-traumatic stress disorder. Despite a gradual improvement of the paranoid symptoms, nightmares, intrusions, and frightfulness, no lasting stabilization could be achieved due to persisting fears of deportation. The case was discussed with the Refugee Law Clinic as well as in the Trauma Network, and the addressed contents were included during further inpatient treatment. The team of the ward reflected on the treatment process in transcultural supervision. With the help of a lawyer, a ban on deportation could finally be achieved. Currently, the patient is undergoing outpatient treatment at the clinic; he has valid residence papers and is receiving additional support from the Psychosocial Center. He is attending a German course, is learning very quickly and intends to start training as a geriatric nurse.

This example shows in a prototypical way the cooperation of different stakeholders of the trauma network for the benefit of the affected person. Different treatment and support services of the network met with a victim who, despite his severe illness, had a high level of commitment and motivation for change. In addition to the patient’s *willingness* to accept the treatment offers, his *ability* to accept the treatment and implement the change in everyday life was also constantly worked on during the treatment. This is a fluid process of empowering the patient more and more to be able to transfer traumatizing events experienced in the past into the personal narrative.

## 5. Implications for Mental Health of Refugees

Trauma-related disorders are complex diseases that can be challenging to treat. Factors such as refugee experiences, sequential traumatization, legal uncertainty with uncertain resident status, and language barriers further complicate the treatment of these disorders and can lead to a significantly increased treatment effort. At the same time, inadequate or lack of treatment leads to an aggravation of individual suffering with the risk of chronification. This can become a significant obstacle to integration, which is consecutively associated with social problems and follow-up costs. A key finding from the research initiated by the trauma network is that the care of refugees requires a two-step approach which should be established as a useful best practice. First of all, in addition to the physical examination, a broad screening for possible mental health risks should take place as early as possible, which could be repeated in the further course. In case of positive findings, further diagnostic measures should then be taken quickly and appropriate treatments initiated to prevent the worsening of the mental health status.

The treatment of refugees requires looking beyond the horizon: social, political, legal, and economic factors mutually influence the mental health of refugees and essentially determine what kind of psychotherapeutic work should be done. In particular, the legal context determines more than just the external framework; it directly influences the associated medical categories of “illness and health.” Improvements in the health status of refugees in the context of inpatient treatment are therefore often only temporary and fragile, because additional negative developments in the asylum process can impair or even reverse improvements that have been made. In the case of traumatized refugees, the interaction of legal and medical aspects must be taken seriously by the medical, legal, and political authorities involved. Residential and asylum law, and the administrative practices involved with them, can dramatically amplify the consequences of the trauma experienced in the country of origin. In essence, this corresponds to “sequential traumatization” [68], which is synonymous with a continuous subjective experience of powerlessness, helplessness, and fear. From a trauma–therapeutic point of view, this is absolutely counterproductive and stands in the way of internally psychologically processing what has been experienced: those affected internalize the structural violence they have experienced, which crystalizes as they continue to experience reverberations of the original trauma, a detrimental process that ensures that the trauma remains salient in the here-and-now.

At this point, however, it should also be pointed out that the effects of legal conditions on health and forms of so-called structural violence do not manifest themselves through anonymous forces that unfold autonomously in a Kafkaesque sense. There are often several options for action within legal requirements, laws, and structures, which can be used by the respective actors as well as the practitioners to the benefit of the refugee. The authors assume that the adequate handling of the special needs of traumatized refugees can only be achieved in a multiprofessional network. Those who face this difficult task alone quickly reach their limits, become helpless and frustrated, or refuse to treat refugees, choosing instead to refer them to “specialized centers”. Cooperation in networks, on the other hand, enables the development of solutions and the use of individual competencies specific to each professional group efficiently and sustainably. Mutual knowledge of existing competencies and direct personal exchange between the actors always ensures that they feel part of a common cause and can face the structural challenges for refugees in the region as a team.

What are the lessons learned from building the trauma network? Starting from individual case discussions, we quickly realized that a network is necessary in order to understand traumatized refugees in their different areas of life from different professional perspectives and to be able to offer individualized help based on this. In this context, it is necessary for individual stakeholders of the network to provide the framework and structure of the network. The supporters of the network contribute their expertise from their respective professional backgrounds. It is possible to rely on the infrastructure and facilities of the institution where the supporters work full-time. Individual facilitators take responsibility for inviting members to regular meetings and for setting up and maintaining a platform for online communication between members of the network. Under this “umbrella”, voluntary help and activity can additionally be brought into the network in a complementary way to one’s own professional background. For many supporters of the trauma network, it is not always possible to make a clear distinction between professional and volunteer activities; rather, there are often fluid transitions between both. Since the network does not currently have its own financial budget beyond the infrastructural use of existing institutional structures, professionals with financial or economic expertise are not currently represented in the network. Through the network, existing professional collaborations between the network’s supporters could be intensified, even beyond the context of flight and trauma. Some of the network’s supporters have their own refugee experience. In the future, it would make sense to recruit more supporters with their own refugee experience for the network, because it seems likely that such experience would increase the ability of the care providers to show greater empathy and understanding to the refugees. An additional active initiation of contact and involvement of official decision-makers, who only made limited use of the opportunity to participate in the network, has proven successful.

On the individual level, patients who sought treatment often appeared rather overwhelmed and helpless at the beginning of treatment. This may result from cultural beliefs but may also be a symptom of the disease itself. Accordingly, patients often expect an active part of the practitioner with clear instructions for their own actions. On the other hand, treatment providers may even be experienced as incompetent if they demand patients to be active and equal partners in treatment planning and implementation, and if therapeutic interventions take the form of questions rather than clear instructions for action. However, if the treatment providers show interest in the prior experiences and presuppositions of those affected with regard to the working methods of psychiatric institutions, including in the country of origin, and see themselves as being learners, mistrust can be reduced, mutual trust can develop, and a fruitful discourse about different treatment concepts as well as empowerment for treatment “at eye level” can emerge.

The development process outlined here can and should be used as a template for other regions and countries—entirely new systems do not need to be erected to care efficiently and compassionately for refugees. Refugees are, of course, not the only vulnerable group for whom attention to syndemic factors is important in overcoming structural health inequities. Health adversities resulting from upstream determinants such as poverty, unemployment, stigma, ethnic minority status, social exclusion, or gender-based violence may, on the one hand, reinforce each other and, on the other hand, promote the occurrence of various somatic and mental diseases, i.e., infectious diseases such as HIV/AIDS, tuberculosis, but also addiction, diabetes, or depression [2,8]. At the same time, protecting particularly vulnerable groups from syndemic suffering for human rights, public health, and health equity reasons is a fundamental social justice issue [7]. Accessible, available, and affordable health care systems that include diverse stakeholders in health care can help to positively influence the underlying social and structural causes of disease rather than the isolated treatment of single symptoms [8].

## 6. Conclusions

Efficient care networks can emerge by rethinking and strengthening existing structures, selectively adding and flexibly adapting them to individual needs and particularities that come with caring for refugees. Accompanying research offers the opportunity to evaluate and further develop one’s own actions and can contribute to an enrichment in the discourse on refugee care, backed by scientific evidence. Thus, over a period of more than 10 years, a structured framework within a scientist–practitioner’s approach has been developed. This framework offers room for individual adjustments and benefits from human cooperation, synergisms, and the cohesive group efforts of a multiprofessional, interdisciplinary network. Recalling one’s roots, traditions, values, such as the universality of human rights, and skills can help to establish the framework for one’s own actions and how to involve others from a syndemic perspective.

## Figures and Tables

**Figure 1 ijerph-19-13436-f001:**
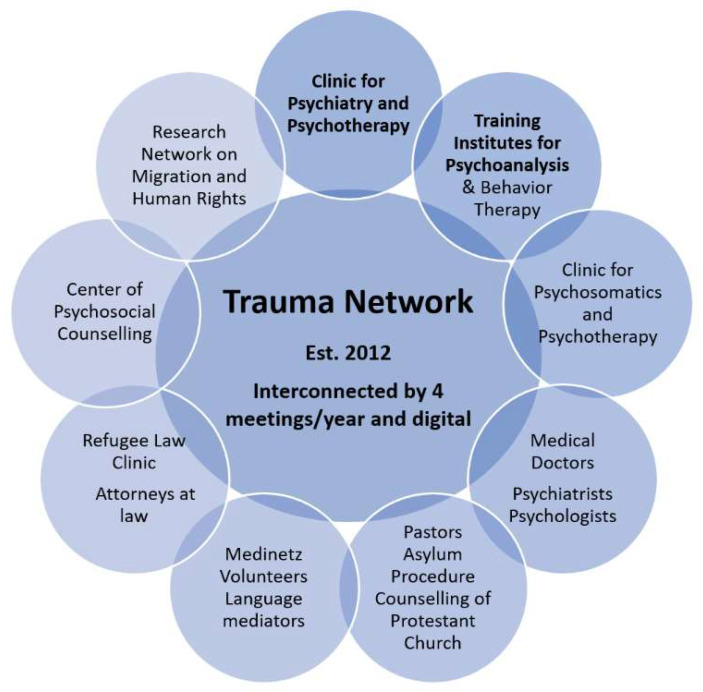
Actors in the trauma network in Middle Hesse for refugee care, hosts bolded. All actors are interconnected and support each other according to need within the framework with their specific competencies (e.g., supervision, legal advice, medical support and treatment, psychotherapy).

## Data Availability

Not applicable.

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
