# Peer review of "Addressing the Mental Health Challenges of Refugees—A Regional Network-Based Approach in Middle Hesse Germany"

_ijerph, 2022, doi:10.3390/ijerph192013436_

Round 1

Reviewer 1 Report

Thank you for the opportunity to review "Addressing the mental health challenges of refugees. A regional network-based approach". The authors provided a conceptual paper on a scientist-practitioner model for preventing trauma and decreasing its negative effects on refugees. The authors demonstrate extensive clinical expertise in this area. This is a critically important and timely topic. The introduction establishes "syndemics" as the theoretical model for the approach and the applicability to the needs of refugees, and provides a strong argument for the approach. Overall the paper is well written and provides many important insights from lessons learned. This approach is a good example for the field. My recommendations are to improve the generalization and replicability. The introduction could be strengthened by including more citations in paragraph 2 to demonstrate convergence with the literature.  The summaries of the different "contributors" are described and provide interesting insights. However, it would be helpful if a figure of the framework were presented to show how the different contributors are linked together with guiding principles that promote the cohesion and continuity between the clinics and centers for refugees.  A figure of the framework might also help the reader better understand the iteration of the scientist-practitioner approach and the benefits. There is considerable research on research-practice partnerships and translational research that can be cited to support this. Can the lessons learned from the collaborators and examples of how they have worked together be further conceptualized and highlighted to promote the generalization and replicability of the framework to the rest of the field?

Author Response

Thank you very much for the valuable recommendations and comments as well as the appreciative feedback. 
We have added to the literature in Section 2 and added a picture of the network. In addition, we have included and referenced literature on other research-practice collaborations. 
In the "Implications for Mental Health" section, we have added a "lessons learned" section.
We hope that our work has been strengthened by this revision, thank you again for the constructive review, and we are looking forward to your valuable feedback.

Reviewer 2 Report

I would like to express my great appreciation for the authors for their hard work and great presentation of the subject. I really enjoyed reading this paper and I also consider this topic as very relevant. I really appreciate that the authors have presented their own perspective about this subject. I have only some small suggestions that the authors might consider to introduce: maybe I missed it and I sincerely appologies for that if I did, but are there any of these type of programmes in Germany or other countries? Could please the authors put short description? Other than that I think the manuscript is excellent and should be accepted! It is also very important - again, my congratulations

Author Response

Thank you very much for your very appreciative feedback, which we were very pleased to receive. We have referred to the question of whether similar other programs exist in Germany.  
We hope that our work has been strengthened by this revision, thank you again for the constructive review and we are looking forward to your valuable feedback.

Reviewer 3 Report

This paper presented an interdisciplinary network of organizations and structures that aims to serve the mental health needs of refugees in Giessen, Germany. It also presented an overview of research findings from several studies related to this network.  This paper is important in that it provided useful description of a real-world collaborative network that serves and examines the complex needs of the refugee population. It is generally well written. However, perhaps due to the large scope of service and research work it attempted to present, it seems a bit disorganized and some details that I consider important are either unclear or missing. I encourage the authors to address the comments below to strengthen the paper.

MAJOR COMMENTS:

Overall comments:

     -          I appreciate the detailed descriptions of the interdisciplinary and inter-sectoral constituents of the Trauma Network and how each plays a role in meeting the needs of the refugee population. Having said that, I wonder if the authors perceive refugees to be passive recipients of care and services offered to them. Do the authors view refugees to be potential agents of change, and if so, where this view is presented in the paper. Research data on the refugees’ mental health and illnesses, helplessness/hopelessness, and coping strategies are potential evidence of their agency (or lackthereof) in the face of traumatic experiences.  Please elaborate this in the paper.

      -          It seems that professionals with financial or economic expertise are not represented in the Trauma Network. Is that correct? Please clarify this in the manuscript.

P. 4, “Establishing the practitioner nexus"

      -          P. 4, “Establishing the practitioner nexus”, line 185-186:  The authors described that the Trauma Network is “open for participation” and that “membership is not formally tied to any obligations or conditions”. Can the authors relate this arrangement to the literature on volunteerism and altruism, and comment on how the open and voluntary nature of membership may affect the effectiveness, ethics, and sustainability of the Trauma Network? If any relevant data have been collected, please report them.

      -          P. 4, “Establishing the practitioner nexus”, line 192-200: The authors described that some people who have had “their own refugee experience" served as interpreters and language/cultural mediators. Did any other members of the Trauma Network have former refugee experience? It seems likely that such experience would increase the ability of the care provider to show greater empathy and understanding to the refugees.  If any relevant data have been collected, please report them.     

     -          Please add a paragraph to describe how the Trauma Network managed power differentials between the refugees and the service providers (the many different groups under the Trauma Network), especially when refugees were enrolled as participants in scientific collaborations (p. 5, lines 209-213) and human research subject protection measures (e.g. in epidemiological data were collected, p. 5, line 299) became necessary.

P. 8, “The Scientific Part: Research”

      -          P. 8, “The Scientific Part: Research”, lines 382-395: This paragraph suggested that research was done, or could have been done, with both regular refugees and “vulnerable refugees” (line 393-395).  Can the authors clarify which types of refugees were recruited into their studies?

     -          Reporting of the studies in this section (pp. 8 to 10) was unclear. Based on my read, 4 studies conducted by the Trauma Network were reported-- references 28 and 29, as well as the ongoing trial comparing the stress of children in families that fled to Germany with those who did not (p. 9, line 454) and the joint project RE-LATER (p. 9, line 459). Please add 1-2 sentences in the introduction of this section (before the paragraph on “Identifying common trauma-related disorders”) to outline the studies to be reported. If the latter 2 studies are registered in any research registries such as Clinicaltrial.gov, please provide the registration numbers.

      -          Please state if all reported studies received approvals from ethics committees as required, as well as describe the particular challenges involved in conducting research with refugees and ways the authors overcame the challenges.

Pp. 11-12 Implications and Conclusions

     -          P.11, line 566-567: Is this Trauma Network the only network of its kind? If it is not, please cite the other networks.  This would allow the readers to better assess if this Network can indeed be used as “a template for other regions and countries” as the authors claimed.

MINOR COMMENTS:

- Please add a diagram to illustrate the organizational structure of the Trauma Network. This would make it easier to understand the relationships among its members.

- For clearer organization of the paper, it may be better to have clear subheadings to distinguish the part that describes the services from the part that describes the scientific research organized by the Trauma Network. It seems the research starts on p. 8 with “The Scientific Part: Research”. Please add a subheading of the same level before the description of services starts.

- P. 10, line 468: Suggest to replace “following subsection” with “next subsection”.

- Pp. 10-11, Case example: Please add 1-2 sentences to explain what the authors intended to illustrate with this case example.

- Please check and fix punctuation errors, word choice issues, and typos. For example:

P. 8, lines 364- 365: the open quotation mark before “Asylum law cases…”

P. 8, line 393: suggest to replace “disabled people” with “people with disability”

Author Response

Thank you very much for the valuable recommendations and detailed comments as well as the appreciative feedback. 
We have added some sections to our manuscript in response to the comments, which can be tracked in the change mode. We thereby tried to adress appropriately to all of your comments.
For example, we added a section on the relationship between providers and patients and commented on helplessness resulting from traumatic experiences. 
We commented on the question of whether a financial expert is part of the network, as well as on the importance of one's own refuge experience among network supporters, in the section "Implications for Mental Health." 
In the section "Establishing the practitioner nexus", we added on the importance of altruism.
We discussed measures of study participant safety and ethics committee opinions in the section "The Scientific Part: Research" and included the study registration number where available.  
We have referred to the question of whether similar other programs exist in Germany and added a diagram. 
For better structuring of the content, we have adapted the font of the headings. 
The mentioned language notes have been implemented.  
We hope that we could strengthen our work by this revision, thank you again for the constructive review. We are looking forward to your valued feedback.